# The Species-Specific Acquisition and Diversification of a K1-like Family of Killer Toxins in Budding Yeasts of the Saccharomycotina

Lance R. Fredericks[1], Mark D. Lee[1], Angela M. Crabtree[1], Josephine M. Boyer[1], Emily A. Kizer[1], Nathan T. Taggart[1], Cooper R. Roslund[1], Samuel S. Hunter[2¤], Courtney B. Kennedy[1], Cody G. Willmore[1], Nova M. Tebbe[1], Jade S. Harris[1], Sarah N. Brocke[1], Paul A. Rowley[1]*

1 Department of Biological Sciences, University of Idaho, Moscow, Idaho, United States of America, 2 iBEST Genomics Core, University of Idaho, Moscow, Idaho, United States of America

¤ Current address: University of California Davis Genome Center, University of California Davis, Davis, California, United States of America
* prowley@uidaho.edu

**Data Availability Statement:** All sequencing data files are available from the NIH genbank database (accession number: MW248137 and Bioproject

## Abstract

Killer toxins are extracellular antifungal proteins that are produced by a wide variety of fungi, including *Saccharomyces* yeasts. Although many *Saccharomyces* killer toxins have been previously identified, their evolutionary origins remain uncertain given that many of these genes have been mobilized by double-stranded RNA (dsRNA) viruses. A survey of yeasts from the *Saccharomyces* genus has identified a novel killer toxin with a unique spectrum of activity produced by *Saccharomyces paradoxus*. The expression of this killer toxin is associated with the presence of a dsRNA totivirus and a satellite dsRNA. Genetic sequencing of the satellite dsRNA confirmed that it encodes a killer toxin with homology to the canonical ionophoric K1 toxin from *Saccharomyces cerevisiae* and has been named K1-like (K1L). Genomic homologs of K1L were identified in six non-*Saccharomyces* yeast species of the Saccharomycotina subphylum, predominantly in subtelomeric regions of the genome. When ectopically expressed in *S. cerevisiae* from cloned cDNAs, both K1L and its homologs can inhibit the growth of competing yeast species, confirming the discovery of a family of biologically active K1-like killer toxins. The sporadic distribution of these genes supports their acquisition by horizontal gene transfer followed by diversification. The phylogenetic relationship between K1L and its genomic homologs suggests a common ancestry and gene flow via dsRNAs and DNAs across taxonomic divisions. This appears to enable the acquisition of a diverse arsenal of killer toxins by different yeast species for potential use in niche competition.

accession number: PRJNA679840). MW248137: https://www.ncbi.nlm.nih.gov/nuccore?term= MW248137 PRJNA679840: https://www.ncbi.nlm. nih.gov/bioproject/679840.

**Funding:** The research was supported by funds provided to PAR by the Institute for Modeling Collaboration and Innovation at the University of Idaho (NIH grant #P20GM104420), the Idaho INBRE Program, an Institutional Development Award (IDeA) from the National Institute of General Medical Sciences of the National Institutes of Health under Grant #P20GM103408, and the National Science Foundation Division of Molecular and Cellular Biosciences grant number 1818368. This work was also supported in part by NIH COBRE Phase III grant #P30GM103324. Funding was also provided by the Office of Undergraduate Research (LRF, SNB, EAK, JSH, and JMB) and the Brian and Gayle Hill Undergraduate Research Fellowship (NMT and CBK) at the University of Idaho. Publication of this article was funded by the University of Idaho Open Access Publishing Fund. The funders had no role in study design, data collection and analysis, decision to publish, or preparation of the manuscript.

**Competing interests:** The authors have declared that no competing interests exist.

## Author summary

Antifungal killer toxins produced by *Saccharomyces* yeasts can be found mostly encoded by cytoplasmic double-stranded RNAs (dsRNAs) rather than the DNA genome. A survey of *Saccharomyces* yeasts has identified a new dsRNA-encoded killer toxin that has a unique antifungal activity and is related in structure and function to the canonical K1 killer toxin. This "K1-like" (K1L) killer toxin was identified to be part of a larger family of DNA-encoded "K1 killer toxin-like" (*KKT*) genes in more distantly related yeasts. *KKT* genes encode active killer toxins and appear to have been acquired sporadically during the evolution of each yeast species, with evidence of ongoing gene duplication. The common ancestry of K1L and homologous killer toxins suggests the transfer of these genes via dsRNAs and DNAs between different yeast species that inhabit similar natural and artificial environments. The potential evolutionary advantage of acquiring killer toxins for niche competition rationalizes the ongoing acquisition and diversification of these genes in yeasts.

## Introduction

Many different species of fungi have been observed to produce proteinaceous killer toxins that inhibit the growth of competing fungal species [1–10]. The killer phenotype was reported in the budding yeast *Saccharomyces cerevisiae* in 1962, when Bevan *et al.* observed that spent culture medium had antifungal properties [11]. The potential future application of killer toxins as novel fungicides has led to the discovery of many different killer yeasts with varying toxicities and specificities [12,13]. In the *Saccharomyces* yeasts, including commonly used laboratory strains, it is estimated that 3–10% are able to produce killer toxins [7–10]. Despite the number of known killer yeasts that have been identified, a complete understanding of the diversity of killer toxins and their evolutionary history is lacking, even within the model yeast *S. cerevisiae*. The horizontal transfer of genetic material between different species of yeast is well appreciated [14–16] and the acquisition of killer toxins can be associated with the horizontal transfer of DNAs between related species that occupy similar ecological niches [17,18]. There is also evidence to support the sporadic genomic capture of killer toxins that are encoded by DNAs and RNAs that are derived from plasmids, viruses, and satellites [19,20]. These examples suggest that mobilization of killer toxins can occur by different mechanisms and that extrachromosomal elements play a role in gene flow between species.

In general, killer toxin production by *S. cerevisiae* is most often enabled by infection with double-stranded RNA (dsRNA) totiviruses of the family *Totiviridae* [21–23]. Totiviruses that infect *Saccharomyces* yeasts are approximately 4.6 kbp in length and encode two proteins, Gag and Gag-Pol (the latter by a programmed -1 frameshift). These proteins are essential for the assembly of viral particles and the replication of viral RNAs [24,25]. Totiviruses act as helper viruses for the replication and encapsidation of 'M' satellite dsRNAs, which often encode killer toxins. These satellite dsRNAs are not limited to *Saccharomyces* yeasts, as they have been identified within other yeasts of the phylum Ascomycota (i.e. *Zygosaccharomyces bailii*, *Torulaspora delbrueckii*, and *Hanseniaspora uvarum* [26–28]) and the phylum Basidiomycota (*Ustilago maydis* and *Malassezia sympodialis* [29–31]). In the Ascomycota, the organization of dsRNA satellites is similar, with all encoding a 5' terminal sequence motif with the consensus of $G(A)_{4-6}$, one or more central homopolymeric adenine (poly(A)) tracts, and a 3' untranslated region (UTR) containing packaging and replication *cis*-acting elements [24]. In all known satellite dsRNAs, killer toxin genes are positioned upstream of the central poly(A) tract and

encode a single open reading frame. Identifying and characterizing dsRNAs is challenging, as the sequencing of dsRNAs currently requires specialized techniques for nucleic acid purification and conversion to cDNAs [32–34]. This has limited our understanding of the diversity of dsRNA-encoded killer toxins within fungi.

In *Saccharomyces* yeasts, there are eight known satellite dsRNA-encoded killer toxins that have been identified (K1, K2, K28, Klus, K21/K66, K45, K62, and K74), with the majority found in the species *S. paradoxus* [35–39]. Owing to their early identification and distinct mechanisms of action, most functional studies have focused on the *S. cerevisiae* killer toxins K1 and K28. *Saccharomyces*-associated killer toxin genes appear to be evolutionarily diverse and unrelated by nucleotide and amino acid sequence. Despite the lack of homology, there are similarities in the posttranslational modifications that occur during killer toxin maturation prior to extracellular export of the active toxin. Each killer toxin is expressed as a pre-processed toxin (preprotoxins) with hydrophobic signal peptides that are required for extracellular secretion [40]. These signal peptides are cleaved by a signal peptidase complex in the endoplasmic reticulum. In the case of K1 and K28 toxins, the resulting protoxins are glycosylated and then crosslinked by disulfide bonds in the endoplasmic reticulum. These disulfide bonds are critical for both stability and toxicity [41–43]. The disulfide-linked protoxins are further cleaved by carboxypeptidases in the Golgi network to yield mature toxins that are secreted via exocytosis [44]. The mature K1 and K28 toxins can be described as α/β heterodimers that are linked by interchain disulfide bonds. Once outside of the producer cell, mature killer toxins can exert their antifungal activities upon competing fungi.

The K1 toxin was the first to be discovered in *S. cerevisiae* and the mechanism of action has been studied extensively (reviewed in [45]). K1 is an ionophoric toxin that attacks the cell membrane of susceptible yeast cells and is mechanistically similar to the K2 toxin [46]. Interaction of K1 with a susceptible cell occurs in a two-step process that involves the initial energy-independent binding of the toxin to the β-1,6-D-glucan polysaccharide of the yeast cell wall [47]. After binding surface glucans, K1 translocates to the cell membrane where it interacts with a secondary receptor, Kre1p [48]. Although there is still uncertainty on the exact mechanism of action of K1, it is likely that intoxication is caused by α-domain dependent formation of membrane channels and the subsequent selective leakage of monovalent cations from the cytoplasm [49]. Importantly, killer toxin immunity is provided by the immature protoxin by a poorly understood mechanism. For K1, this activity has been mapped to the α-subunit and 31 amino acids of the adjacent γ-subunit, which is usually removed during toxin maturation [50,51].

This manuscript describes two novel findings related to killer toxin biology. The first is the identification of a novel killer toxin encoded by a satellite dsRNA in *S. paradoxus* by screening a large collection of *Saccharomyces* yeasts. This killer toxin has low sequence identity to the canonical K1 toxin produced by *S. cerevisiae* but has a similar secondary structure and domain organization. Due to its relatedness to K1, we have named this new killer toxin K1-like (K1L). The second major finding is that this is the first example of a dsRNA-encoded killer toxin from *Saccharomyces* yeasts that has significant homology to DNA-encoded killer toxins that we have named "K1 killer toxin-like" (*KKT*) within multiple species of the Saccharomycotina. These proteins represent a family of K1-like killer toxins that are both diverse and show possible signs of rapid protein evolution. This work provides insights into the expansion and horizontal transfer of killer toxin genes in yeasts, whether they are encoded by DNAs or mobilized and replicated as dsRNAs by viruses.

## Methods

### Killer phenotype assays

Killer toxin production by yeasts was measured using killer yeast agar plates (yeast extract, peptone, and dextrose (YPD) agar plates with 0.003% w/v methylene blue buffered at pH 4.6),

as described previously [32]. Toxin production was identified by either a zone of growth inhibition or methylene blue-staining of the susceptible lawn yeasts. The pH optima of killer toxins produced by different yeasts was measured on killer yeast agar plates adjusted to pH values of 4.0, 4.5, 5.0 and 5.5.

### Killer toxin enrichment

Strains of killer yeast were grown in 2 mL of YPD medium (pH 4.6) overnight at room temperature with vigorous shaking (250 rpm). The culture was centrifuged at 3,100 × g for 5 min followed by filtration of the culture medium through a 0.22 μm filter. Filtered growth medium was added 1:1 with 4°C supersaturated ammonium sulfate solution and mixed by inversion and incubated on ice for 3 h. The precipitated proteins were collected by centrifugation at 20,800 × g for 10 min at 4°C. The supernatant was then removed, and the precipitated proteins suspended in 10 μL of YPD pH 4.6. Killer toxins were incubated either at room temperature, or heat-inactivating at 98°C for 2 min before treating lawns of susceptible yeasts spread onto killer assay agar plates.

### RNA extraction and purification

Double-stranded RNAs were extracted as described previously [32], with the following modifications: The extracted dsRNAs were not incubated with oligo d(T)25 magnetic beads and 2× LTE buffer was replaced with 2× STE (500 mM NaCl; 20 mM Tris-HCl, pH 8.0; 30 mM EDTA, pH 8.0). Single-stranded RNAs were extracted from yeasts using the RNeasy RNA extraction kit (Qiagen) with the bead beating method and an on column DNase I digestion. After column elution, an additional incubation with DNase I (Omega Bio-Tek) was performed at 37°C for 10 min followed by 75°C for 10 min when the removal of additional contaminating DNAs was required.

### Curing *Saccharomyces* yeasts of dsRNAs

Cycloheximide, anisomycin, or high temperatures were used to create strains of yeasts that lacked satellite dsRNAs. Yeasts were first grown overnight in 2 mL of liquid YPD medium before 1 μL of cell suspension was transferred to YPD agar with either cycloheximide (0.4–5.0 μM) or anisomycin (0.8 μM). Yeast cultures were incubated for 2–5 days at 23°C to recover surviving cells. Curing satellite dsRNAs using temperature involved incubating yeast cultures on YPD agar for 2–5 days at 30°C, 37°C, or 40°C. Growing cells were streaked onto YPD agar plates from the heat-treated agar plates and were incubated for an additional 2–3 days at 23°C. The colonies resulting from chemical or temperature treatment were analyzed for the loss of killer toxin production by replica plating onto killer assay agar plates seeded with a killer toxin sensitive yeast strain.

### Short read sequencing of satellite dsRNAs

The preparation of dsRNAs, cDNAs, Nextera Illumina libraries, and sequence analysis was the same as previously reported, with several amendments detailed below [32]. Reads were cleaned with fastp or HTStream (https://github.com/ibest/HTStream) and *de novo* contigs were assembled with SPAdes assembler v3.11.1 using default parameters [52,53].

### Sanger sequencing of SpV-M1L

Reverse transcriptase PCR (RT-PCR) was used to generate overlapping DNAs that represented the genetic sequence of the satellite dsRNA named SpV-M1L. The approximate molecular

weight of these DNAs was determined by agarose gel electrophoresis and capillary electrophoresis (Fragment Analyzer, Agilent Technologies Inc). DNAs were cloned using the pCR-Blunt II-TOPO vector and subjected to Sanger sequencing. 5' and 3' RACE were used to determine the terminal ends of the dsRNA molecules using the protocol provided by the manufacturer (Invitrogen) (see S4 File for the full list of primers).

### Cloning of genome-encoded killer toxin genes

Genomic DNAs were extracted from *K. africana*, *N. dairenensis*, *N. castellii*, *T. phaffii*, and *P. membranifaciens* using the method of Hoffman and Wilson (1987) and were used as templates for PCR (see S4 File for the full list of primers) [54]. Killer toxin genes were cloned into pCR8 by TOPO-TA cloning (Thermo Fisher) and the DNA sequences were confirmed by Sanger sequencing. The K1L gene was commercially synthesized (GeneArt by Thermo Fisher) and used as a PCR template to amplify K1L. The PCR-derived K1L gene was cloned into pCR8 by TOPO-TA cloning and confirmed via Sanger sequencing. All killer toxin genes were subcloned using Gateway technology into the destination vector pAG426-Gal-ccdB for ectopic expression in either *S. cerevisiae* or *S. paradoxus* (see S5 file for all plasmid sequences) [55,56].

### Expression of K1L and homologs from the Ascomycota

For ectopic expression of killer toxins, plasmids encoding toxin genes were used to transform either *S. cerevisiae* BY4741 or a non-flocculant derivative of *S. paradoxus* A12 (named A12C) [57]. Transformants were selected on complete medium lacking uracil. To assay toxin expression, a single colony of each transformed strain was used to inoculate a series of consecutive overnight cultures in 2 mL of complete medium lacking uracil, first with dextrose, then raffinose, and finally galactose at 30°C with shaking (250 rpm). The optical density of the final 2 mL culture was normalized to an $OD_{600}$ of 1 and 1 mL was centrifuged at $3,000 \times g$ for 5 min. The supernatant was removed, and the cell pellet was disrupted by gentle agitation. 2.5 μL of the resulting cell slurry was used to inoculate YPD and YPG plates (with 0.003% w/v methylene blue, pH 4.6) seeded with a killer toxin-susceptible yeast strain. Inoculated plates were incubated for 48–72 h at 25°C until killer toxin production was visible.

### Phylogenetic analyses

Killer toxin gene sequences were aligned using MUSCLE and manually trimmed to represent the most confident alignment of the α-domain. MEGA (version 7) was used for phylogenetic analysis using neighbor-joining and maximum likelihood methodologies. The optimal model for amino acid substitution was determined as the Whelan and Goldman model with a gamma distribution. 500 bootstrap replicates were used to construct a phylogenetic model with the highest log-likelihood.

## Results

### The identification of new strains of killer toxin-producing yeasts

A total of 110 strains of *Saccharomyces* yeasts were obtained from the USDA Agricultural Research Service (ARS) culture collection and screened to identify the production of novel killer toxins. The first screen used eight yeasts from four different species to indicate toxin production on "killer assay medium", which is a complex YPD medium buffered to pH 4.6, the optimal pH for most *Saccharomyces* killer toxins. Methylene blue was also added to the medium as a redox indicator of cell death. This assay found that 22% (24/110) of yeasts could inhibit the growth of at least one strain of yeast (S1 File). To identify the types of killer toxins

based on their unique spectrum of activities, 13 killer yeasts were further screened against 46 indicator lawn yeasts. Four strains of *S. cerevisiae* that have been previously described to produce killer toxins of unknown types were also included (NCYC1001, NCYC190, CYC1058, and CYC1113) [58–60]. To facilitate the classification of different toxin types, yeasts that produce canonical K1, K28, and K74 killer toxins, and a non-killer yeast were included for comparison (Fig 1). The degree of growth inhibition by each killer yeast was scored qualitatively based on the appearance of zones of growth inhibition and methylene blue staining of the surrounding indicator strain on agar plates (Fig 1 and S2 File).

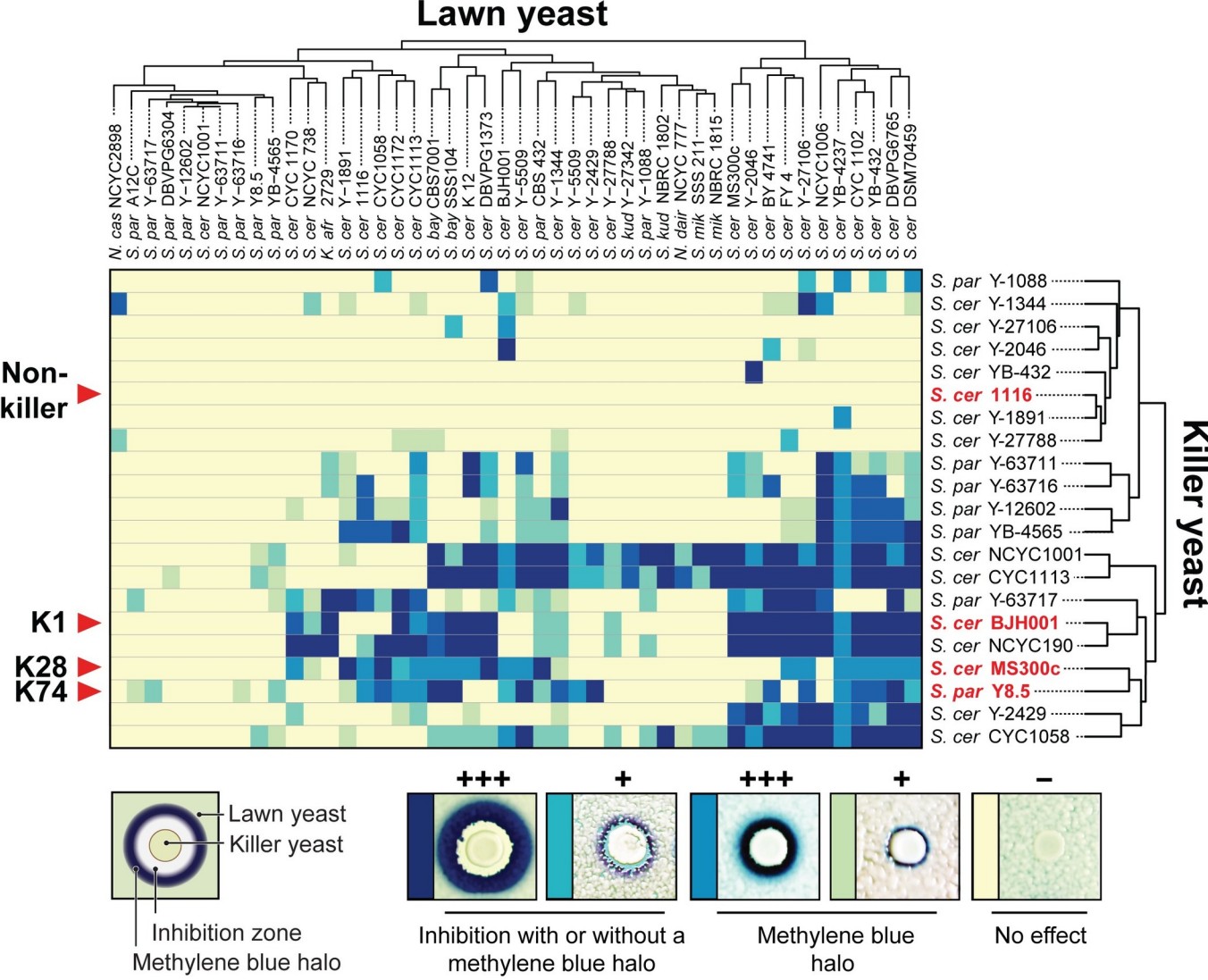

**Fig 1. The strain- and species-specificity of killer toxins produced by *Saccharomyces* yeasts.** A total of 21 killer yeasts were assayed for killer toxin production on agar plates seeded with 46 different lawn yeasts. Yeast from the USDA (ARS) culture collection are designated by either 'Y' or 'YB'. Killer toxin activity was qualitatively assessed based on the presence and size of zones of growth inhibition and/or methylene blue staining around killer yeasts as diagrammed (bottom left). Darker colors on the cluster diagram represent a more prominent killer phenotype with yellow indicating no detectable killer phenotype. The non-killer yeast strain *S. cerevisiae* 1116 was used as a negative control (bottom right). Results were analyzed using the R package gplots to cluster killer yeasts and lawn yeasts based on killer toxin production and susceptibility, respectively. *S. cer*—*S. cerevisiae*, *S. par*–*S. paradoxus*, *S. bay*–*S. bayanus*, *S. kud*–*S. kudriavzevii*, *S. mik*–*S. mikatae*, *N. dair*–*N. dairenensis*, *N. cas*–*N. castellii*, and *K. afr*–*K. africana*.

The known strain- and species-specificity of killer toxins have been previously used to bio-type different yeasts [61–65] and to identify groups of unknown killer toxins [3,66]. Therefore, cluster analysis was used to group killer yeasts based on their ability to inhibit growth of lawn strains and revealed that no two strains had the exact same spectrum of antifungal activity. *S. cerevisiae* CYC1113, CYC1058, and NCYC1001 had the broadest antifungal activities but did not cluster with any of the known killer toxins. Clustering found that that the antifungal speci-ficity of *S. cerevisiae* NCYC190 was most closely correlated with that of a K1 killer yeast, with 85% identical interactions with competing lawn strains. The killer toxin activity of *S. para-doxus* Y-63717 also clustered with K1 (50% identical interactions), showing both gain and loss of activity. This result was particularly intriguing because evidence of K1 production by *S. paradoxus* is inconsistent between different research groups with recent studies suggesting that K1 toxins are unique to *S. cerevisiae* [8,67,68]. The remaining killer yeasts were identified to have weaker inhibitory activities with no clustering with K1, K28, or K74 killer yeasts.

## Killer yeasts harbor satellite dsRNAs that encode proteinaceous killer toxins

The production of killer toxins by yeasts is often associated with the presence of satellite dsRNAs that are maintained by totiviruses [25,45]. To determine whether satellite dsRNAs were responsible for the observed killer phenotypes, killer yeasts were treated with either cyclo-heximide, anisomycin, or incubated at elevated temperatures to select for the loss of killer toxin production [69,70]. The majority of killer yeast (12/19) lost their killer phenotype after exposure to chemical or thermal insult (Fig 2A). The same treatments were unable to select for the loss of the killer phenotype in the remaining seven strains, which suggests that their killer toxins were genome-encoded (S1 Fig). To determine if the killer phenotype was correlated with the presence of dsRNAs, cellulose chromatography was used to selectively purify dsRNAs from the killer yeasts [32]. The analysis of extracted dsRNAs revealed that the loss of the killer phenotype was 100% correlated with the loss of satellite dsRNAs (Fig 2A).

The dsRNA-encoded killer toxins identified have a pH optimum mostly between 4.5 and 5, with no inhibitory activity at pH 5.5 (Fig 2B). To confirm that the identified killer toxins are proteinaceous, each was purified by ammonium sulfate precipitation and used to challenge susceptible yeasts. Zones of inhibition were clearly visible on confluent lawns of yeast cells for all of the killer toxins tested but not for a non-killer yeast or a strain cured of its dsRNA satellite (Fig 2C). The inhibitory activities of these precipitates were heat-labile, and the toxicity was lost after incubation at 98˚C for 2 minutes. The identified killer toxins have similar biochemi-cal characteristics to known proteinaceous killer toxins despite their differing inhibitory effects towards yeasts.

## The discovery of a new killer toxin produced by S. paradoxus

To identify the unknown killer toxins produced by killer yeasts, dsRNAs were purified and subjected to a short-read sequencing pipeline for dsRNAs [32]. BLASTn analysis of *de novo* assembled contigs revealed that the satellite dsRNAs within strains CYC1058 and NCYC1001 encode canonical K2 toxins and NCYC190 a canonical K1 toxin (S2 Fig and S1 Table). The sequence reads derived from the dsRNAs of Y-63717 assembled into 125 different contigs, with six >750 bp in length and a coverage score >1,000 (Fig 3A). BLASTn analysis of these high-quality contigs identified the dsRNA genome of the totivirus L-A-45 from *S. paradoxus* N-45 with 100% coverage and 95.5% nucleotide identity [36]. However, the remaining contigs did not match the nucleotide sequence of any known *Saccharomyces* yeast killer toxin or dsRNA satellite. Short-read sequencing was augmented with a combination of 5' and 3' RACE,

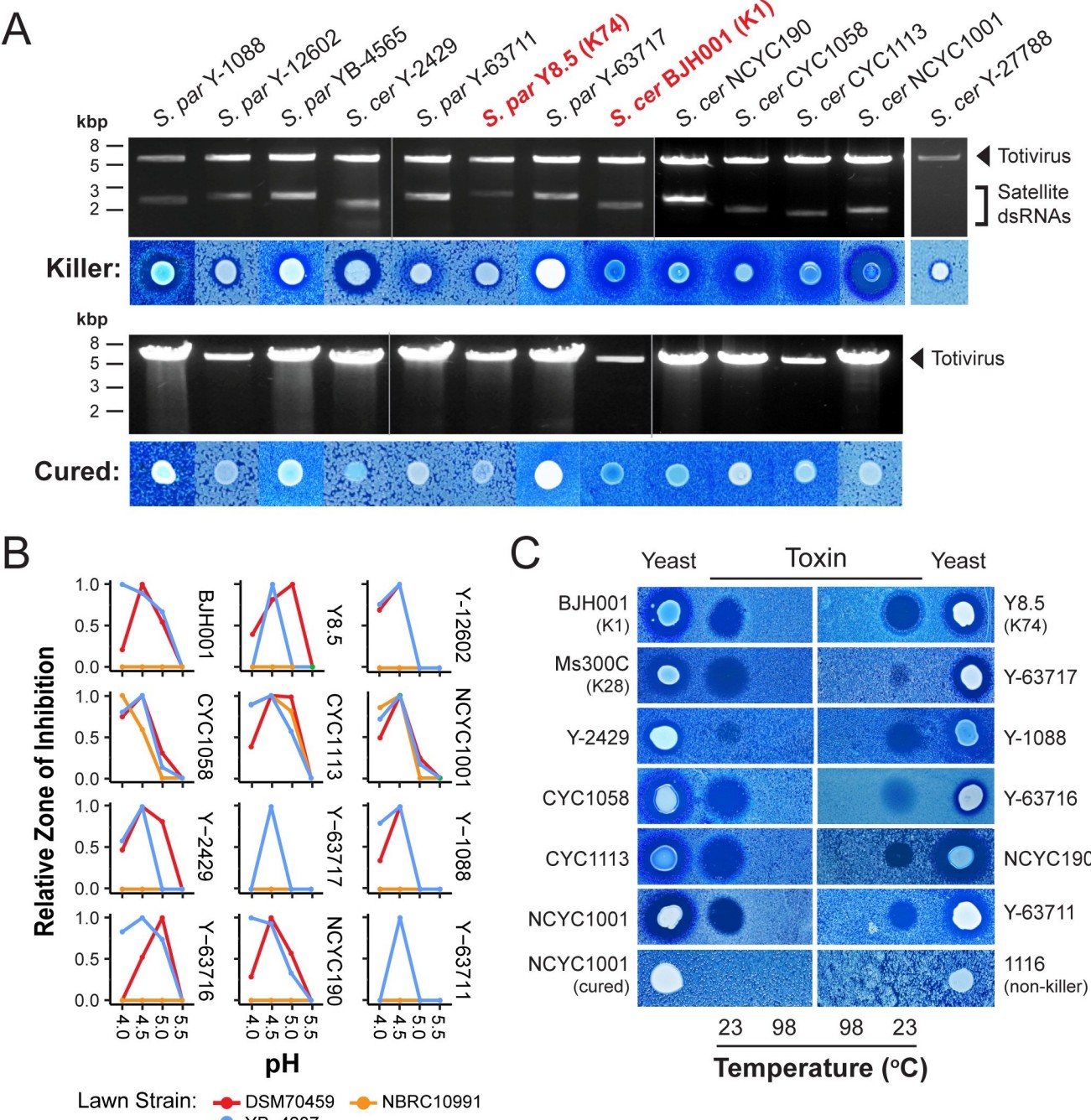

**Fig 2. Analysis of dsRNAs present within killer yeasts and the properties of their killer toxins.** (A) The extraction and analysis of dsRNAs from totiviruses and associated satellites in *Saccharomyces* yeasts that produced killer toxins (top panel) or lost the killer phenotype after exposure to cycloheximide, anisomycin, or elevated temperatures (bottom panel). *S. cerevisiae* Y-27788 is included as a representative killer yeast that lacks a satellite dsRNA. (B) The relative pH optimum of killer toxins against three different indicator lawn yeasts *S. cerevisiae* YB-4237, DSM70459, and NBRC10991 using YPD buffered to pH 4.0, 4.5, 5.0, and 5.5. (C) Enrichment and concentration of killer toxins from spent growth medium by ammonium sulfate precipitation and their loss of inhibitory activity after incubation at 98˚C.

RT-PCR, Sanger sequencing, and capillary electrophoresis to assemble the complete sequence of the dsRNA satellite from Y-63717 (Figs 3B and S3). The novel satellite dsRNA is approximately 2,371 bp in length with a single open reading frame (ORF) that encodes a protein of

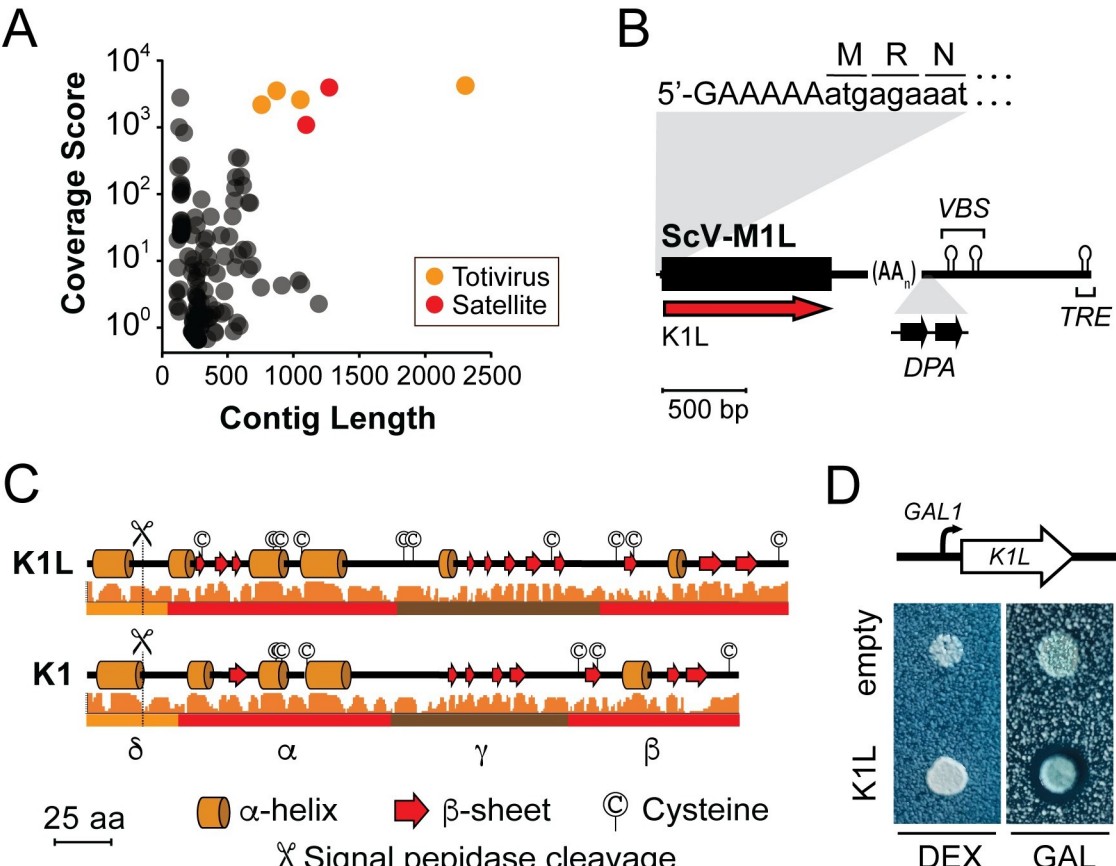

**Fig 3. Short-read sequencing and analysis of the K1L killer toxin from *S. paradoxus* Y-63717.** (A) Sequence contigs after *de novo* assembly of sequence reads represented by contig coverage score and contig length. BLASTx analysis was used to annotate contigs as similar to totiviruses or satellite dsRNAs. (B) Schematic of the organization of the SpV-M1L satellite dsRNA (C) Jpred secondary structure prediction of the K1 and K1L killer toxins with confidence score plotted as a histogram. Predicted domain boundaries are drawn below the secondary structure prediction [76]. Signal peptidase cleavage site were predicted by SignalP and TargetP (S3 File). (D) Ectopic expression of K1L from a plasmid in the non-killer yeast *S. paradoxus* A12 induced by the presence of galactose in the growth medium.

340 amino acids. The 5' ORF is positioned upstream of a central poly(A) tract of ~220 bp (Figs 3B and S3). The 5' terminus has a nucleotide sequence of 5'-GAAAAA that is found in many satellite dsRNAs (Figs 3B and S3) and is predicted to fold into a large stem-loop structure (S4 Fig). Downstream of the poly(A) tract in the 3' UTR there are elements of secondary structure that are indicative of replication (terminal recognition element; *TRE*) and packaging signals (viral binding site; *VBS*) that have been well characterized in the canonical M1 satellite dsRNA from *S. cerevisiae* (ScV-M1) (Figs 3B and S4) [71–73]. In addition to RNA secondary structures, there are also two direct repeats of the sequence motif named "Downstream of Poly(A)" (*DPA*; 5'-CTCACCYTGAGNHTAACTGG-3') that is found in different satellite dsRNAs isolated from *S. paradoxus* (M45, M74, and M62), *S. cerevisiae* (M1 and Mlus), *Zygosaccharomyces bailii* (MZb), and *Torulaspora delbrueckii* (Mbarr) (Figs 3B and S3) [27,74].

The length and positioning of the 5' ORF of the satellite dsRNA in *S. paradoxus* Y-63717 suggested that it encoded a killer toxin (Fig 3B). A PSI-BLAST search of the NCBI database with two iterations found that this putative killer toxin had weak homology to the canonical K1 toxin from *S. cerevisiae* (99% coverage, 18% amino acid identity, e-value $4 \times 10^{-17}$) (Fig 4A and S2 Table). Based on this homology, the putative killer toxin was named K1L (K1-like) and

the dsRNA satellite was named Saccharomyces paradoxus virus M1-like (SpV-M1L). The organization of the functional domains of K1L appears to be similar to K1 based on the secondary structure, conserved cysteine residues, and the predicted signal peptidase cleavage sites (Fig 3C) [75]. K1L contains ten cysteine residues, two of which are likely important for interchain disulfide linkage (Cys91) and killer toxin immunity (Cys257) based on their alignment with cysteines from K1 [41]. To confirm that the K1L is an active killer toxin, it was ectopically expressed by the non-killer strain *S. paradoxus* A12C using a galactose inducible promoter. A well-defined zone of growth inhibition was visible when the strain was grown on galactose-containing medium (Fig 3D). No K1L toxin expression was observed when cells were plated on dextrose-containing growth medium. Together, these data confirm the identification of a new dsRNA satellite in *S. paradoxus* and a novel killer toxin related to K1.

## K1L homologs are found in yeasts of the Saccharomycotina

The PSI-BLAST search that identified K1L as a homolog of K1 also identified 24 hypothetical "K1-like Killer Toxin" (*KKT*) genes in six diverse species of non-*Saccharomyces* yeasts from the subphylum Saccharomycotina (Figs 4A and S5). The species *Kazachstania africana*, *Naumovozyma castellii*, *Naumovozyma dairenensis*, *Tetrapisispora phaffii*, and *Pichia membranifaciens* encode *KKT* genes that are homologous to K1L (aligned >300 amino acids, e-value <$10^{-80}$) and represent yeasts from the families *Saccharomycetaceae* and *Pichiaceae* (Fig 4A and S3 Table) [77]. Importantly, genomic *KKT* genes appear to be unique to these particular species and absent from other related yeasts (i.e. *Kazachstania naganishii*, *Tetrapisispora blattae*, and *Pichia kudriavzevii*). The length of all Kkt proteins are between 153–390 amino acids with 11 being similar in length (340 amino acids) (Fig 4A) and amino acid identity (16–28%) to K1L (S2 Table). In addition, BLASTn was used to identify 14 additional pseudogenes that, in some species, outnumber intact *KKT* genes (Fig 4B and S3 Table). All *KKT* genes and related pseudogenes are found in multiple copies that vary in frequency between different yeast species and are mostly located within subtelomeric regions (within ~20 kb of the assembled chromosome ends) (Fig 4B). Of the 38 *KKT* genes and pseudogenes found within six different species, only two are positioned away from the subtelomeric regions in the yeasts *N. dairenensis* and *N. castellii* (Fig 4B). Analysis of the chromosomal position of these two genes revealed that their insertions are unique to each species, absent from other related species at the syntenic chromosomal location, and inserted close to tRNAs (S6 Fig). It was noted that six of the *KKT* genes and pseudogenes from *K. africana* contained a characteristic "GAAAAA" sequence motif close to the start codon of each ORF, similar to K1L and other killer toxins (S6 Fig).

To ascertain the evolutionary history and relatedness of K1L and its homologs, a multiple amino acid sequence alignment was constructed. The most confident alignment was achieved between the putative α-domain of each protein and included eight truncated proteins with premature stop codons. Phylogenetic analysis was performed using maximum likelihood (Fig 4C) and neighbor-joining (S7 Fig) methodologies with 500 bootstrap iterations as implemented by MEGA [78]. The tree topologies of the two phylogenetic models are in general agreement and show the distinct clustering of killer toxins from each species (Figs 4C and S7). Importantly, K1L is most closely related to the Kkt1 proteins from *T. delbrueckii* and *P. membranifaciens*. The Kkt proteins from *K. africana* form a monophyletic clade that contrasts those from *T. phaffii* and *N. dairenensis* that appear to be polyphyletic. There also appears to be some support for a common ancestry of killer toxins within the *Naumovozyma* yeasts. BLASTn analysis and nucleotide alignment of the 5' and 3' UTRs of the *KKT* genes from *K. africana* indicates that flanking nucleotide sequence is between 83–94% identical over ~2,000

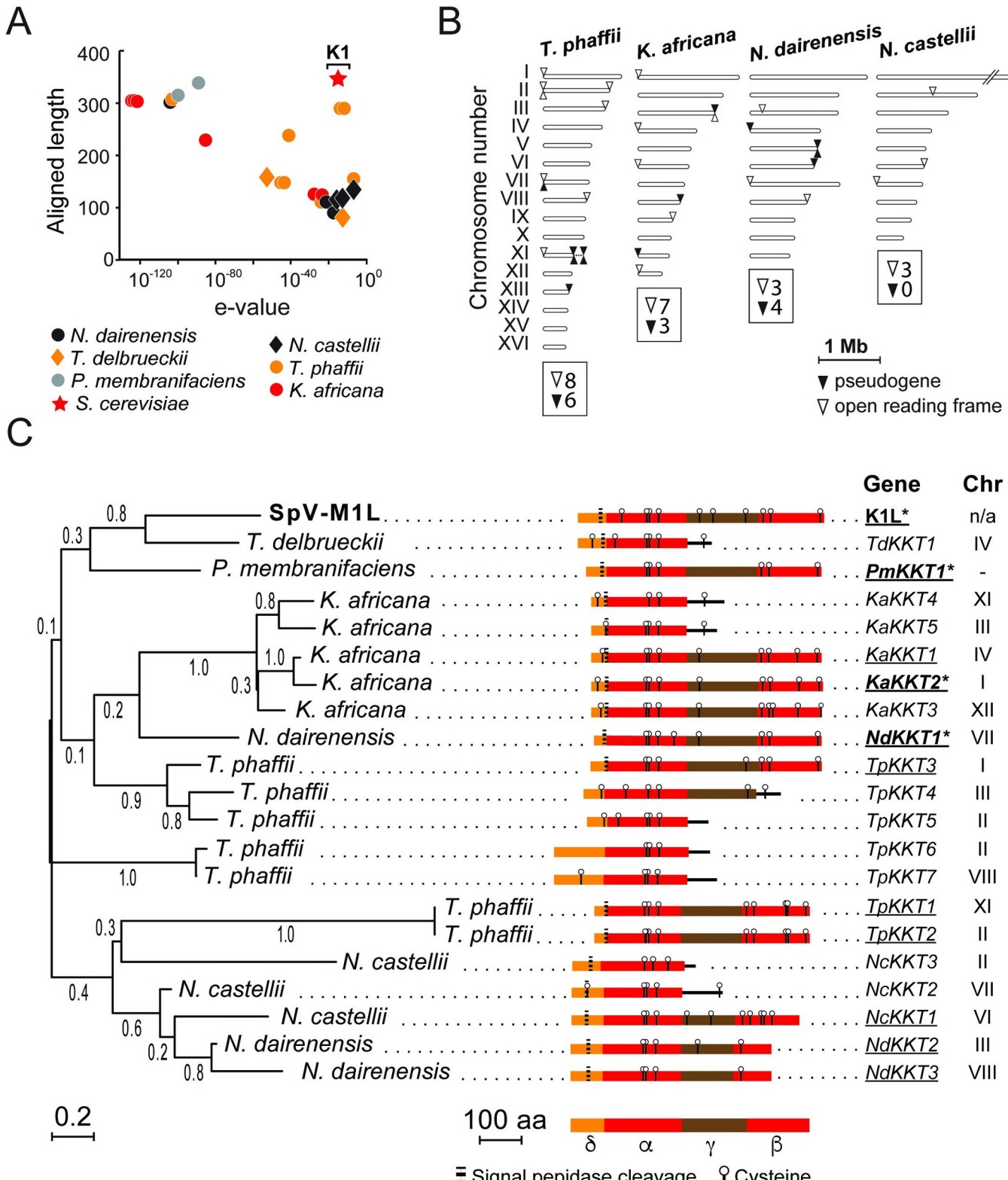

**Fig 4. *KKT* genes are distributed across the subphylum Saccharomycotina and are primarily subtelomeric in their genomic location.** (A) PSI-BLAST analysis of the killer toxin encoded upon the satellite dsRNA from *S. paradoxus* Y-63717. (B) Location of *KKT* genes and pseudogenes on the linear representation of the chromosomes of four yeasts of the *Saccharomycetaceae*. Chromosome I of *N. castellii* did not contain any *KKT* genes and was truncated in the diagram due to its large size. (C) Unrooted maximum likelihood phylogeny of the aligned α-domain of 21 killer toxins from six species of yeast and one

dsRNA satellite. Numerical values represent the bootstrap support for the placement of each node. The domain organization of each protein is illustrated and annotated based on the four-domain structure of K1. Signal peptidase cleavage site predicted by SignalP and TargetP (S3 File). Underlined killer toxin names represent those that were cloned and functionally tested for antifungal activity. *Killer toxins with confirmed antifungal activities.

bp, suggesting gene duplication. There is also evidence of gene duplication of *KKT* genes within *N. dairenensis* based on their close phylogenetic relationships and similar untranslated regions. To ascertain the evolutionary trajectory of *KKT* paralogs, the rate of accumulation of nonsynonymous (dN) and synonymous (dS) mutations were calculated for *KKT* paralogs that could be confidently aligned by their nucleotide sequences. When dN/dS was calculated for all codons over the evolutionary history of the sequence pairs, all three pairs of paralogs have evolved under purifying selection since their duplication (dN/dS = 0.72, 0.41, 0.23) [79]. A domain-resolution approach using a sliding window to calculate dN/dS supports the predominance of purifying selection with a few localized peaks of possible weak positive selection (dN/dS >1) in the α- and/or γ-domains (S8 Fig).

## A new family of K1-like antifungal killer toxins

The relatedness of the *KKT* genes to the killer toxin K1L suggests that they encode antifungal killer toxins. To assay for killer toxin production by the yeasts that encode *KKT* genes, *P. membranifaciens*, *N. dairenensis*, *N. castellii*, *T. delbrueckii*, *T. phaffii*, and *K. africana* were used to challenge 19 different yeast on killer assay plates. With the exception of *N. castellii*, all of the *KKT*-encoding yeast species produced killer toxins that caused growth inhibition of at least one other yeast (Fig 5A and 5B). Each of these killer yeasts was also immune to its own killer toxin, but susceptible to those produced by other *KKT*-encoding yeasts. The production of killer toxins by these species is consistent with the previously reported killer activity of *P. membranifaciens*, *T. delbrueckii*, and *T. phaffii* [27,80,81]. There was no evidence of satellite dsRNAs in any of the *KKT*-encoding yeasts, except for the detection of an unknown high molecular weight dsRNA within *P. membranifaciens* NCYC333 (S9 Fig). The differences in killer toxin production by five strains of *P. membranifaciens* suggested that there could be strain-specific polymorphisms in *KKT* genes (S10 Fig). The published genome sequence of *P. membranifaciens* Y-2026 revealed a large central deletion in the γ-domain of its *KKT* gene (S10 Fig). Sanger sequencing of the same *KKT* gene from *P. membranifaciens* Y-2026 acquired directly from the NRRL culture collection failed to identify the same deletion, instead there was an indel within the γ-domain that caused the truncation of the killer toxin gene (S10 Fig). Sequencing of the *KKT* gene from *P. membranifaciens* NCYC333 confirmed a full-length gene that correlated with robust killer toxin production by the strain. However, *P. membranifaciens* Y-2026 was still able to express killer toxins, suggesting the production of other alternative antifungal molecules.

Although *P. membranifaciens*, *N. dairenensis*, *T. delbrueckii*, *T. phaffii*, and *K. africana* are killer yeasts (Fig 5A and 5B), it was unclear whether *KKT* genes were directly responsible for the observed production of killer toxins. Indeed, *T. phaffii* has been reported to express an antifungal glucanase and K2 killer toxin-related genes have been found in the genome of *K. africana* [74,80]. To confirm the expression of *KKT* genes in killer yeasts, total genomic DNA and RNA were extracted from six species of killer yeasts and used as templates for PCR and RT-PCR with primers targeting ten different *KKT* genes (S11 Fig). Due to the similarity of the *KKT* genes it was not possible to unambiguously distinguish between transcripts from *TpKKT1* and *TpKKT2*, or *KaKKT1* and *KaKKT2*. Of the six species assayed, *KKT* RNA transcripts were detected in *T. phaffii*, *N. dairenensis*, and *K. africana* indicating active gene transcription under laboratory conditions (Fig 5C). To demonstrate that *KKT* genes are active

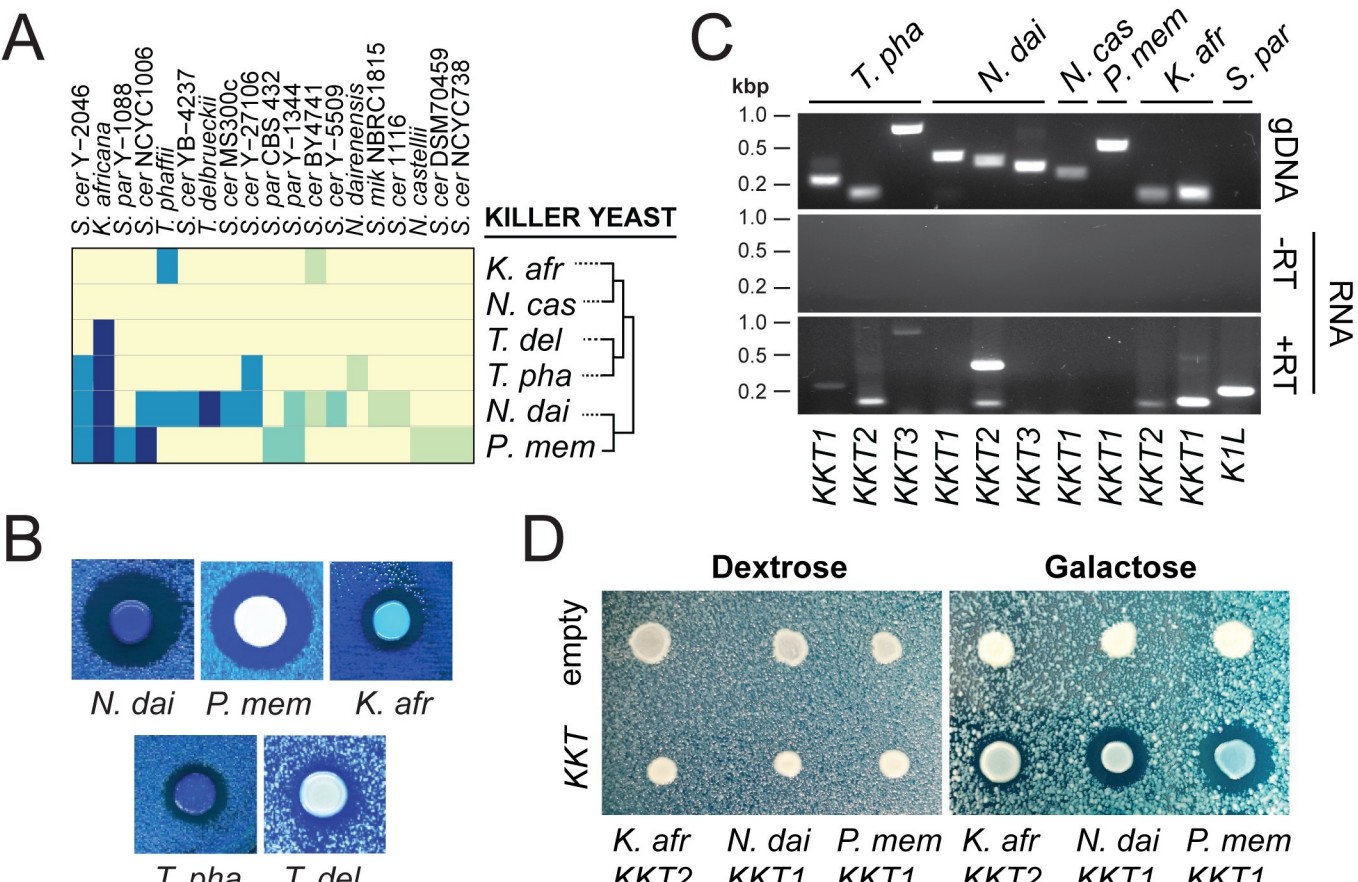

**Fig 5. The antifungal activity of Kkt toxins from *K. africana*, *N. dairenensis*, and *P. membranifaciens*.** (A) *KKT*-encoding yeasts were assayed for killer toxin production on killer assay agar plates seeded with indicator strains. Killer toxin production was judged by the presence of zones of growth inhibition or methylene blue staining (see the key in Fig 1). (B) Representative pictures of killer toxin production by different species of yeast from the Saccharomycotina. (C) PCR detection of *KKT* gene DNA and RT-PCR detection of RNA transcripts under laboratory conditions. *S. paradoxus* Y-63717 served as a dsRNA control. (D) Galactose-dependent ectopic expression of *KKT* genes from *K. africana*, *P. membranifaciens* and *N. dairenensis* can inhibit the growth of *K. africana*. *N. dai–N. dairenensis*, *N. cas–N. castellii*, *K. afr–K. africana*, *P. mem–P. membranifaciens*, *T. pha–T. phaffii*, *S. cer–S. cerevisiae*, *S. mik–S. mikatae*, *S. par–S. paradoxus*, and *T. del–T. delbrueckii*.

killer toxins with antifungal activities, a subset was cloned into a galactose inducible expression vector (labelled in Fig 4C). Active killer toxin production from a non-killer strain of *S. cerevisiae* transformed with *KKT* genes was assayed against eight lawns of yeasts using galactose containing agar plates. The majority of *KKT* genes did not cause any noticeable growth inhibition or methylene blue staining of competing yeasts when expressed from *S. cerevisiae* (S12 Fig). However, killer toxins from *P. membranifaciens* NCYC333 (*PmKKT1*), *K. africana* (*KaKKT2*), and *N. dairenensis* (*NdKKT1*) were able to create visible zones of growth inhibition (Fig 5D). No growth inhibition was observed when gene expression was repressed by plating cells on dextrose. Altogether, these data show that *KKT* genes are transcriptionally active and encode active killer toxins. This confirms the discovery of a new family of K1-like genome-encoded antifungal proteins in the Saccharomycotina.

## Discussion

The most significant finding of this study is the discovery of a novel satellite dsRNA that encodes a killer toxin related to K1 and a larger family of DNA-encoded homologs in

Saccharomycotina yeasts. The relatedness of killer toxins encoded on dsRNAs and DNAs suggests that the origins of K1L are outside of the *Saccharomyces* genus, with killer toxin gene mobilization and interspecific transfer by dsRNAs. These killer toxins have been shown to be biologically active and are diverse in their amino acid sequences with evidence of their rapid evolution by gene duplication and elevated rates of non-synonymous mutations. This demonstrates the likely benefits of killer toxin acquisition and the ongoing mobilization of these genes between divergent species of yeasts. The more specific implications of our findings are discussed below.

## Horizontal acquisition and copy number expansion of killer toxins in fungi

*KKT* genes have most likely been acquired by horizontal gene transfer because of their sporadic distribution and lack of common ancestry in closely related yeast species. Moreover, the lack of relatedness of *KKT* genes in these species suggests that they were acquired independently. Fungi are known to acquire foreign DNAs from other species of fungi [14–18,82–84] as well as bacteria [14,85,86]. The interspecific capture of DNAs derived from retrotransposons, viruses, and plasmids has also been observed [19,20,87,88]. Specifically, genome integrated copies of dsRNA-encoded killer toxins homologous to KP4, K1, K2, Klus, and Kbarr have been found within bacteria and fungi, indicating gene flow between dsRNAs and DNAs across taxonomic divisions [74,89]. However, the vast majority of these putative killer toxins are uncharacterized, and it remains unclear as to whether they are biologically active.

Phylogenetic evidence suggests that cross-species transmission of viruses between fungi has occurred on multiple independent occasions [90,91]. Laboratory experiments have also successfully demonstrated extracellular [92,93] and interspecific virus transfer [36]. In particular for *Saccharomyces* yeasts, mating and hybridization between different species has been observed, and is a mechanism for gene introgression, as well as for the acquisition of retrotransposons and plasmids. The close association of many yeast species in natural and anthropic habitats may increase the likelihood of horizontal gene transfer or invasion by dsRNA viruses and satellites [94–96]. Specifically, the satellite dsRNA that was identified in this study (named SpV-M1L) and an unrelated satellite dsRNA (SpV-M45) are both found within sympatric Far Eastern yeast strains of *S. paradoxus* [36,97]. The parasitism of L-A-45 by both of these satellite dsRNAs in different strains of yeast strongly implies that they were acquired by horizontal gene transfer. The apparent structural similarities of the satellite dsRNA-encoded *VBS* elements suggest that, despite the evolutionary distance between SpV-M1L and SpV-M45, each has evolved a similar mechanism to enable the hijack of the replication and packaging machinery from the L-A-45 totivirus. Unlike *Saccharomyces* yeasts, the presence of active RNAi within the *KKT*-encoding yeasts would hinder the horizontal acquisition of dsRNA viruses from other yeasts, which could explain the abundance of genome-encoded killer toxins and paucity of dsRNA viruses and satellites (S5 Fig) [98,99]. However, *T. delbrueckii* can support the replication of the putative totivirus TdV-LAbarr and an associated satellite [27], which suggests that this strain has lost its RNAi machinery or the totivirus has the ability to evade or disrupt RNAi similar to other mycoviruses [100–102]. The apparent invasion of these yeasts by viruses could enable the acquisition of killer toxins encoded upon satellite dsRNAs.

To capture killer toxin genes from dsRNAs, the erroneous reverse transcription of mRNAs by endogenous retrotransposons would allow their insertion into a yeast genome by the retrotransposon integrase protein [103]. The ancestral *KKT* gene in *K. africana* could have been captured directly from dsRNAs as all extant paralogs encode a 5'UTR "GAAAAA" motif, which is characteristic of satellite dsRNAs. The 5' UTR is often conserved during mRNA capture by retrotransposons [103]. While the majority of *KKT* genes are subtelomeric in their location, we

have observed that non-subtelomeric *KKT* genes from *N. dairenensis* and *N. castellii* have been uniquely inserted into genomic loci near tRNA genes. The genomic integration of retrotransposon cDNAs is selectively targeted to tRNA genes and many extrachromosomal nucleic acids are identified at loci adjacent to tRNAs [19]. This means the potential mobilization of these genes by retrotransposons and direct insertion by integrase or cellular DNA repair mechanisms [104]

## The potential benefits of killer toxin acquisition

Acquisition of foreign nucleotide sequences can be associated with adaptation to a specific environmental niche, including genes associated with nutrient acquisition, virulence, the stress response, and interference competition (e.g. allelopathy) [18,19,84]. Killer toxin-like genes have been identified within the genomes of many different species of fungi and could represent selection for functional killer toxins to improve competitive fitness. Indeed, the production of killer toxins by different species of yeasts has been consistently shown to provide a competitive advantage, particularly in a spatially structured environment at an optimal pH for killer toxin activity [105–110]. However, competition between different killer yeasts likely selects for locally adapted populations that are immune to the predominant killer toxins in a specific environmental niche [6]. The laboratory evolution of killer yeast populations has also shown that killer toxin exposure increases the prevalence of killer toxin resistance [111]. High levels of killer toxin resistance perhaps account for the low prevalence of killer yeasts within natural yeast populations and that killer toxin production might not always be advantageous. However, the rise of killer toxin resistance within a population could perhaps drive the acquisition of new killer toxins or the subfunctionalization of existing killer toxins to maintain a selective advantage. In the context of the multicopy dsRNA satellites and expanded killer toxin gene families in yeasts, it would seem logical to assume that natural variation would provide a pool of genetic diversity on which natural selection could act.

The majority of *KKT* genes are found to have been acquired within the subtelomeric regions of chromosomes that would facilitate gene expansion due to elevated rates of homologous recombination between telomeric repeat sequences. As has been noted for the subtelomeric *MAL* gene family, gene duplication enables evolutionary innovation that is also evident in *KKT* paralogs by possible weak signatures of positive selection [112]. Other genome-encoded fungal killer toxin families have also undergone copy number expansion and are experiencing elevated rates of non-synonymous substitutions [89,113]. Both KP4-like and Zymocin-like killer toxin families appear to have roles during antagonistic interactions with plants and fungi, respectively, which could drive the continued evolution of novel killer toxins. In addition to the expansion of *KKT* genes in yeasts, there is also gene loss and pseudogenization. *KKT* gene inactivation is biased towards truncations that leave the α-domain and a small portion of the γ-domain. This same α/γ region of K1 is the minimal sequence required for functional killer toxin immunity that could provide a selective advantage by protecting yeasts from exogenous killer toxins related to *KKT* [50,51]. *KKT* yeasts all encode C-terminally truncated *KKT* genes and are mostly resistant to other *KKT* killer yeasts. However, despite encoding several truncated *KKT* genes, *K. africana* appears to be naturally susceptible to many of the killer toxins produced by other *KKT* encoding yeasts, including its own killer toxin when ectopically expressed by *S. cerevisiae*. Subtelomeric Sir-dependent gene silencing could account for this apparent susceptibility under laboratory conditions [114,115]. This would prevent the expression of *KKT* genes and associated immunity by *K. africana*, although transcripts from full-length *KKT* genes were detected in *K. africana*.

K1L and its homologs represent a unique example of the mobilization and subtelomeric expansion of DNA- and dsRNA-encoded killer toxin genes in different species of yeasts. The

diversity of dsRNAs and the known prevalence of killer yeasts suggests that more killer toxins await future discovery and characterization. This will enable a better understanding of their role in fungal biology and provide greater insights into the mechanisms of horizontal gene transfer in eukaryotes.

## Supporting information

**S1 Fig. Double-stranded RNAs extracted from killer yeasts that retained their killer pheno-type after exposure to cyclohexamide (CHX).** Agarose gel electrophoresis was used to show dsRNAs present in killer yeasts. Satellite dsRNAs are labeled as dsRNAs that are smaller than the associated totivirus dsRNAs. The high molecular weight DNAs are predicted to be mito-chondrial in origin.
(TIF)

**S2 Fig. Coverage and contig lengths of dsRNAs from different strains of killer yeasts.** The scatter plots represent all contigs generated after *de novo* assembly of sequence reads. M satel-lites are labeled in red according to their relatedness to other previously described sequences.
(TIF)

**S3 Fig. Sequence analysis of the dsRNA satellite M1L from *S. paradoxus* Y-63717.** (A) Cov-erage of the assembled sequence reads for M1L and the positioning of the expected products from three RT-PCR reactions to amplify portions of the *K1L* ORF (reaction 1), the 3' UTR (reaction 2), and across the internal poly(A) tract (reaction 3). PCR products shown by agarose gel electrophoresis and their estimated sizes as determined from fragment analysis are repre-sented. (B) The positioning of the repeated *DPA* element is represented relative to the genomes of eight dsRNA satellites. (C) The consensus sequence derived from 15 *DPA* elements is shown as a sequence logo and multisequence alignment.
(TIF)

**S4 Fig. RNA secondary structure predictions of M1L, M1, and M45 (+) strand 5' and 3' ter-mini.** (A) Secondary structure prediction of the 5' terminal structures. Start codons for the translation of preprotoxin synthesis are highlighted by a grey outline. Numbers represent nucleotides from the 5' terminus. (B) Putative replication signal represented as a stem-loop at the 3' end of M1L and M1 satellite. Numbers represent distance from the 3' terminal nucleo-tide. (C) Putative viral binding sites (*VBS*) with a 5' facing 'A' bulge present in the stem-loops (indicated by an arrow). Numbers represent distance from the 3' terminal nucleotide. The mFold server was used to calculate the change in free energy for each structure [116].
(TIF)

**S5 Fig. Cladogram of selected budding yeasts of the Saccharomycotina.** The panel details the presence or absence of totiviruses, satellite dsRNAs, the killer phenotype, *KKT* genes, and RNAi within 15 species. '?' denotes the uncertainty of the putative dsRNA virus detected in *P. membranifaciens* in S8 Fig. WGD indicates the ancestral yeast species that underwent a whole genome duplication.
(TIF)

**S6 Fig. The unique genomic insertion of *KKT* genes in *N. dairenensis* and *N. castellii*.** (A) *N. dairenensis KKT2* inserted into chromosome III and *N. castellii KKT3* inserted into chro-mosome II. Genes flanking *KKT* insertions are represented as black arrows and demonstrate synteny between related genomes. Single red triangles represent tRNA genes. Broken lines rep-resent gaps in synteny. (B) 5' UTR sequence from *KKT* genes and one pseudogene identified

within *K. africana*.
(TIF)

**S7 Fig. Phylogenetic model of the evolutionary relationship between Kkt proteins using the neighbor-joining method.** Unrooted neighbor-joining phylogeny of the aligned α-domain of 21 *KKT* proteins from six species of yeast and K1L from one dsRNA satellite (SpV-M1L). Numerical values represent the bootstrap support for the placement of each node.
(TIF)

**S8 Fig. Paralogous *KKT* genes are potentially evolving under weak positive selection in different species of yeasts.** Three sliding window dN/dS calculations are shown for the comparison of three pairs of closely related genes in three yeast species. The x-axis represents the nucleotide (nt) number of each gene and is shown in the context of the predicted domain organization of each pair of genes. Omega values represent the whole gene dN/dS value for each gene pair. Sliding windows analysis was performed using a window of 90 nucleotides with a 30 nucleotide overlap. Asterisks in the *T. phaffii* dN/dS plot represent instances where dS = 0 and dN > 0.
(TIF)

**S9 Fig. The absence of satellite dsRNAs within killer yeasts of the Saccharomycotina.** Agarose gel electrophoresis of dsRNAs extracted from different killer yeasts. Stained bands in lane 2 represent a canonical totivirus and satellite dsRNA from *S. cerevisiae*.
(TIF)

**S10 Fig. Strain-specific production of killer toxins by *P. membranifaciens*.** (A) Killer toxin production and partial purification from *P. membranifaciens*. (B) Mutations within *KKT1* in the context of the proteins secondary structure organization (as predicted by Jpred) from strain Y-2026 compared to a full-length active killer toxin sequenced from strain NCYC333. Arrows represent β-sheets and cylinders represent α-helices. "c" represents cysteine residues.
(TIF)

**S11 Fig. The specificity of primers targeting *KKT* genes.** Where possible, primers pairs were designed to recognize different full-length *KKT* paralogs from *T. phaffii*, *N. dairenensis*, and *K. africana*. DNA plasmids were used as templates to test their specificity. Due the similarities in DNA sequences, specific primer pairs were not identified for *TpKKT1*, *TpKKT2*, and *KaKKT2*.
(TIF)

**S12 Fig. Ectopic expression of *KKT* genes causes the inhibition of *K. africana* and is dependent on the induction of *KKT* expression.** Galactose-dependent ectopic expression of *KKT* genes from a multicopy plasmid by *S. cerevisiae* on agar plates seeded with different species of yeasts. Key: 1. pUI114 (*NcKKT1*), 2. pUI109 (*TpKKT2*), 3. pUI110 (*TpKKT1*), 4-pUI111 (*TpKKT3*), 5. pUI112 (*NdKKT3*), 6. pUI113 (*NdKKT2*), 7. pML115 (*NdKKT1*), 8. pML117 (*KaKKT2*), 9. pML118 (*KaKKT1*), 10. pML116 (*PmKKT1*), Nc. *N. castellii* NCYC2898, Nd. *N. dairenensis* NCYC777, Tp. *T. phaffii* Y-8282, Pm. *P. membranifaciens* NCYC333.
(TIF)

**S1 Table. Nucleotide and amino acid identity of killer toxins from different strains of *S. cerevisiae* compared to canonical K1 and K2.**
(DOCX)

**S2 Table. Amino acid identity and similarity between K1, K1L and Kkt proteins.**
(DOCX)

**S3 Table. Species that encode genomic killer toxins that are homologous to K1 and K1L.**
(DOCX)

**S1 File. A large-scale screen to identify killer yeasts in the *Saccharomyces* genus.**
(PDF)

**S2 File. Image data illustrating the susceptibility of yeast to a selection of potent killer toxins produced by *Saccharomyces* yeasts.**
(DOCX)

**S3 File. SignalP and TargetP predictions for K1, K1L and Kkt proteins.**
(TXT)

**S4 File. Supplementary file listing all primers, plasmids, and yeast strains used in this study.**
(PDF)

**S5 File. Supplementary FASTA file with the DNA sequences of all plasmids used in this study.**
(TXT)

## Acknowledgments

We would like to acknowledge the NRRL Agricultural Research Service culture collection for providing the Rowley laboratory with a diverse collection of *Saccharomyces* yeasts. We would also like to thank Dr. Antonia Santos (Complutense Yeast Collection, Complutense University of Madrid) for the CYC yeasts, Prof. Manfred J. Schmitt (Saarland University, Saarbrücken, Germany) for *S. cerevisiae* MS300c, Dr. Gianni Liti (University of Nice) for the genetically tractable strains of *S. paradoxus* and *S. cerevisiae*. We would also like to acknowledge Dr. Gianni Liti and Dr. Scott Minnich for helpful comments on the draft manuscript. Opinions, findings, and conclusions or recommendations expressed in this material are those of the author(s) and do not necessarily reflect the views of the funders.

## Author Contributions

**Conceptualization:** Emily A. Kizer, Paul A. Rowley.

**Data curation:** Lance R. Fredericks, Mark D. Lee, Samuel S. Hunter, Paul A. Rowley.

**Formal analysis:** Lance R. Fredericks, Mark D. Lee, Angela M. Crabtree, Samuel S. Hunter, Paul A. Rowley.

**Funding acquisition:** Paul A. Rowley.

**Investigation:** Lance R. Fredericks, Mark D. Lee, Angela M. Crabtree, Josephine M. Boyer, Emily A. Kizer, Nathan T. Taggart, Cooper R. Roslund, Courtney B. Kennedy, Cody G. Willmore, Nova M. Tebbe, Jade S. Harris, Sarah N. Brocke.

**Methodology:** Lance R. Fredericks, Angela M. Crabtree, Paul A. Rowley.

**Project administration:** Paul A. Rowley.

**Supervision:** Paul A. Rowley.

**Validation:** Paul A. Rowley.

**Writing – original draft:** Paul A. Rowley.

**Writing – review & editing:** Lance R. Fredericks, Mark D. Lee, Angela M. Crabtree, Josephine M. Boyer, Emily A. Kizer, Paul A. Rowley.

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
