## [Decision Letter · Decision Letter 0]

5 Nov 2020

Dear Dr Rowley,

Thank you very much for submitting your Research Article entitled 'The Species-specific Acquisition and Diversification of a Novel Family of Killer Toxins in Budding Yeasts of the Saccharomycotina.' to PLOS Genetics. Your manuscript was fully evaluated at the editorial level and by independent peer reviewers. The reviewers appreciated the attention to an important topic but identified some aspects of the manuscript that should be improved.

We therefore ask you to modify the manuscript according to the review recommendations before we can consider your manuscript for acceptance. Your revisions should address the specific points made by each reviewer.

[LINK]

Yours sincerely,

Justin C. Fay

Associate Editor

PLOS Genetics

Hua Tang

Section Editor: Natural Variation

PLOS Genetics

Reviewer's Responses to Questions

**Comments to the Authors:**

Reviewer #1: Fredericks et al explore the presence of killer phenotypes and killer toxins in a group of budding yeast species. They found additional strains that produce killer toxins, identified genes encoding killer toxins, and explored the evolutionary history of those genes. Overall, i think the paper makes a significant contribution to our understanding of an interesting topic. My biggest criticism is that I felt the novelty of the gene family was overstated. I appreciate the identified genes are greatly diverged from other K1 genes and have a ‘unique spectrum of activity’, but I wasn’t convinced it is helpful to call them a new family. Based on the title, I anticipated the family would be totally distinct from other identified killers. I think calling the family K1-like genes and including K1 is justified, but claiming they are a novel distinct family is not. I have listed additional suggestions and points of confusion or concern below in an effort to help the authors improve the paper.

Line 153: Sequence reads were deposited to the NCBI Sequence Read Archive with the accession number: TBA.

It would help increase the accessibility of the results to explain the significance of the pH and what’s happening with the methylene blue staining in the first paragraph of the results.

Figure 1: labels for the indicator strains are missing. It is hard to match the ‘k’ labels on the left to the clusters on the right.

Line 250: I had trouble following the numbers here. Were 19 strains tested? If so, how do you get 86% of those strains? 17 is 89% and 16 is 84%. There are 12 strains shown that have dsRNAs that could be cured of both the killing and the dsRNAs. I think i could follow this better if all the observed classes and the numbers in those classes were presented. Were the strains lacking the dsRNAs the only ones that could not be cured of the killer phenotype?

Line 253: “The same treatments were unable to select for the loss of the killer phenotype in five representative strains that lacked dsRNA satellites (Y-1344, Y-2046, YB-432, Y-1891, and Y-27788).” I couldn’t find the data supporting that statement.

Line 255: “Relative to the wild type strains, most of the cured strains exhibited an elevated copy number of totivirus dsRNAs.” Was copy number formally assayed? If so those quantifications should be presented. Are there additional standard that would allow comparing band intensities between gels?

Figure 2: The text says that 19 strains were assayed, but the figure doesn’t present all of them. I found the additional strains in the gel in supplement, but i didn’t see additional data for the curing treatment. It would also be helpful to see a totivirus negative control in A and a nonkiller strain in C.

In 2B, It is difficult to tell which pHs were quantified. Data points in addition to the lines would help. An additional table in the supplement might also help. I can’t distinguish the colors well enough to know which line corresponds to which strain.

Line 277: “BLASTn analysis of de novo assembled contigs revealed that dsRNAs within strains CYC1058 and NCYC1001 encode canonical K2 toxins and NCYC190 a canonical K1 toxin (Fig. S2)” From this, i expected S2 to show an alignment showing sequence similarity, but it plots contig length and coverage scores, which are not good support for the statement that the sequences are of K1 and K2 toxins.

Line 290: The reference to Fig S3 should be to Figure 3B

Line 294: Not clear why Figure S3 is referenced here.

Line 301: Not clear why Figure 2 is referenced here.

Figure S5: Legend says some genes are colored green, but i can only see red and black. There is also a typo “Brocken” in the legend.

Line 369: “The acquisition of KKT genes in T. phaffii appears to have occurred on three separate occasions resulting in three distinct clades of genes that have significant sequence divergence from each other and other KKT genes.” That is one model to explain that pattern, but others are possible. I suggest rephrasing this as a possibility rather than as something that is proven by the data.

“Each clade appears to be composed of 2-3 closely related paralogs, suggesting recent gene duplication events of a single ancestral gene.” It is also possible the duplications were not so recent, but that there is gene conversion between family members.

Positive selection: The number of differences between the compared genes is difficult to determine from Figure S7. I am inclined to suspect the peaks of dN/dS>1 are noise based on few differences, rather than convincing evidence of positive selection. It is not clear how big the sliding windows are or how many changes differentiate the pairs of genes. The key for the symbols on the genes is also missing in S7.

Figure 5: The color code for A is not provided here. Is it the same as the earlier figure? In B) Is T. delb a negative control here showing no killing, or is the phenotype minimal killing? In C, I suggest showing the Gal and glucose control plate together in one figure.

Line 400: “However, Y-2026 was still able to produce killer toxins, suggesting the production of other antifungal molecules by P. membranifaciens.” It also would not be unprecedented that the mutant can still sometimes generate a full length product. Sometimes mutations that should lead to premature truncations can still make some full length protein.

For the KKT genes in genomes, it would be nice to test if they are transcribed under the conditions in which you observed killing. As the authors point out, it is possible other genes are causing the killer phenotypes. Perhaps RNAseq data exists for these species? If not, perhaps do some RT-PCR? This will not prove the genes cause the killer phenotypes, but it could add further support for that notion.

Reviewer #2: Review of, “The Species-specific Acquisition and Diversification of a Novel Family of Killer Toxins

in Budding Yeasts of the Saccharomycotina” by Fredericks et al.

This is an extensive characterization of a large group of ‘killer’ strains of various yeasts, including many determined by dsRNAs and another group determined by chromosomal genes. The completeness of the work and the careful exploring of many aspects of the biology and evolution of these viruses and chromosomal genes makes a very interesting story, and the the work is of quite high quality. I have only a few suggestions for minor modifications.

A. The authors state that the killer genes are “a diverse arsenal of killer toxins for use in niche competition.” Also, a section of the Discussion, “The Benefits of Killer Toxin Acquisition” includes this view. This is certainly a widespread view of the killer systems, but it seems to be contradicted by the low incidence of toxin-producing or even toxin-resistant strains among wild isolates. It has been suggested, originally by Meinardt (?spelling) for the K. lactis DNA plasmid-based killer system, that the toxin is there to prevent the cell from losing the plasmid (or virus in this case), rather a more subversive role of the dsRNAs than the idea of helping the yeast host to conquer the niche world. The “9-10%” frequency of killer toxins is not typical of past studies, some of which have only been 1 or 2 % of isolates. I would suggest that the authors discuss this alternative view as well.

B. Line 468: It is suggested, but not quite clear to me whether you are saying that T. delbrueckii has RNAi and yet can still be infected with dsRNA viruses. If so, please include the fact that T. delbrueckii has an RNAi system.

C. Fig S4: I found the apparent conservation of the architecture of the M ‘putative viral particle binding sites’ quite remarkable, considering the evolutionary distance of the M dsRNAs. I assume this is a result of the ‘helper’ L-A Totiviruses, particularly their Pol domains that bind the site, being more closely related. The authors might want to comment on this aspect.

Reviewer #3: This manuscript describes the diversity of newly discovered K1-like (K1L) toxin-producing genetic elements found throughout the Saccharomycotina. Killer toxins are secreted by fungal cells and kill sensitive fungal cells, often of the same species. The toxin K1, which is expressed on a cytoplasmic dsRNA virus in S. cerevisiae, has been studied for decades, and a few other toxins have been discovered in S. cerevisiae and other fungi. But the diversity and evolution of these toxins are poorly studied, and this manuscript describes observations that give a window onto how toxin-coding genetic elements may evolve within and among fungal groups.

The authors discovered the K1L virus as part of a screen of S. cerevisiae and its sister species S. paradoxus. They found diverse combinations of target yeasts killed by toxin-producing Saccharomyces and diverse optimal pH values for toxins. After dsRNA sequencing, most toxins and associated toxin-coding genetic elements were previously known, but K1L, from a S. paradoxus strain, was previously undescribed (although weakly homologous to K1). A BLAST search found K1L homologs on the nuclear genomes of six other yeast species. Some of these homologs were pseudogenes, but most of the homologs coding for ORFs, when cloned, were able to be expressed in S. cerevisiae and produce toxins capable of killing sensitive yeasts. These newly discovered sequences were found on either nuclear or cytoplasmic nucleic acids, were polyphyletic, and when located on chromosomes were found in diverse and non-syntenic genomic locations (although most often in subtelomeric chromosomal regions). This diversity was taken as evidence for repeated horizontal gene transfer of toxin-coding genetic elements.

While researchers have previously found diverse toxin-coding genetic elements (both in terms of phylogeny and type of nucleic acid), and toxin genes are frequently transmitted horizontally, especially in prokaryotes, to my knowledge the result that the same toxin is likely horizontally transmitted and also coded in different ways among eukaryotic species is new. I also appreciated the new information presented on the general diversity natural history of killer toxins; while K1 is a model system in S. cerevisiae cell biology, the diversity and evolution of killer toxins among fungi is generally poorly understood. The results in this manuscript are surprising because the well-studied dsRNA viruses encoding many Saccharomyces toxins are thought to be only vertically transmitted. While the study doesn’t identify a mechanism of horizontal transmission (there are some possibilities suggested in the discussion), it does produce convincing evidence that this horizontal transmission not only happens at least sometimes, but also that conversion of viral RNA to genomic DNA or vice-versa must be a part of the process. I think the information presented in this manuscript is interesting and new, and should be published in PLoS Genetics. This manuscript is a strong contribution to our understanding of horizontal gene transfer among eukaryotes, and is also a considerable contribution to yeast researchers’ understanding of killer toxin systems.

I have a few minor suggestions for improvements to the manuscript, mostly in terms of manuscript organization and how information is presented.

1) My biggest criticism of this manuscript regards its organization: the main conclusion, that these new K1L toxins are widespread throughout the Saccharomycotina and have a pattern consistent with horizontal transfer of DNA and RNA elements, is buried in the manuscript and minimized in the abstract. The one exception is that these results are prominently presented in the abstract. I understand the need to report results in chronological order, but this strategy doesn’t effectively communicate the relative importance of results. This problem can be resolved with a rewording of the abstract and a refocusing of the introduction. Additionally, would it be possible to better integrate the two “stories” in this manuscript (that killer phenotypes within Saccharomyces are diverse and that K1L shows up in diverse places)?

2) It was sometimes difficult to keep track of the various strains discussed in this manuscript, especially in the Saccharomyces screens early in the manuscript. Would it be possible to give more information about why these particular strains are presented/how they were chosen? Indicating categories of strains on the figures would also help. Additionally, in line 106, the manuscript says that 45 indicator lawn strains were used, but in Figure 1 it says 47. Did I miss the missing two somewhere? This paragraph might benefit from being rewritten to help readers keep track of strains.

3) The text indicates amino acid identity between K1 and K1L, and among K1L and newly discovered K1L-homologous nuclear sequences. But Figure 4a gives e-value and aligned length. I was interested in a plot that shows the aa identities mentioned in the text; could one be included, perhaps as an additional supplementary figure?

Other comments:

Lines 228-229: Could which known killers be indicated?

Lines 369-371: This statement might need to be weakened a little bit because the bootstrap values on the tree in Figure 4C are very low for some of these clades.

Lines 444-445: I’m not sure I quite understood what was meant by “independent origins”. Independent HGT events? Something else?

Line 457: Yeasts in general or Saccharomyces in particular?

Lines 476-478: This sentence was difficult to understand. Could it be reworded?

Figure 2: It doesn’t look like this figure represents all of the killers identified in Figure 1. Why not/why were these chosen for the figure?

Figure S1: What does “n/a” indicate? An explanation of the DNA band would be helpful in this figure.

A comparison of non-Saccharomyces killer sequence homology to K1L vs. homology to K1 would be a welcome addition to the data presented in this manuscript.

It might be nice to include a species tree of the species investigated (and the species mentioned on lines 331-333) so readers can easily compare phylogenetic patterns between K1L sequences and host species.

I found a few typos:

Line 17: space in the word “these”

Lines 65-66: “in Saccharomyces yeasts” is repeated

Line 167: I couldn’t find these primers in table S1. Was Table S4 meant?

Figure S5 legend: What is colored green in this figure (this might be a pdf conversion issue)? Also, “Broken” is misspelled

Figure S10 legend: Part A seems to be missing

**Have all data underlying the figures and results presented in the manuscript been provided?**

Reviewer #1: **No: **I listed a few places in the comments to authors where i could not find data to support statements. The SRA number was also listed as TBA in the methods.

Reviewer #2: Yes

Reviewer #3: Yes

PLOS authors have the option to publish the peer review history of their article (what does this mean?). If published, this will include your full peer review and any attached files.

Reviewer #1: No

Reviewer #2: No

Reviewer #3: No

---

## [Decision Letter · Decision Letter 1]

21 Dec 2020

Dear Dr Rowley,

Thank you very much for submitting your Research Article entitled 'The Species-specific Acquisition and Diversification of a Family of K1-like Killer Toxins in Budding Yeasts of the Saccharomycotina.' to PLOS Genetics.

The manuscript was fully evaluated at the editorial level and by independent peer reviewers. The reviewers appreciated the attention to an important topic but identified some concerns that we ask you address in a revised manuscript

We therefore ask you to modify the manuscript according to the review recommendations. Your revisions should address the specific points made by each reviewer.

[LINK]

Yours sincerely,

Justin C. Fay

Associate Editor

PLOS Genetics

Hua Tang

Section Editor: Natural Variation

PLOS Genetics

Reviewer's Responses to Questions

**Comments to the Authors:**

Reviewer #1: The paper is improved by the revisions, but there are still two claims that are not sufficiently supported by data. These should be removed.

First, starting the assay with the same number of cells is insufficient to claim that there is a higher copy number of the totivirus after curing in Figure 2. I can see that the bands are brighter in the figure, but the bands we are comparing are on different gels without a common standard. I would have more confidence if it was clear if the data were all part of the same prep, assayed in the same PCR, and then run on the same gel.

The second claim is that there is positive selection. I appreciate that the authors added the size of the sliding window in their analyses. The authors have not, however, presented the number of synonymous and nonsynonymous differences between the compared proteins. Without seeing those numbers, i remain concerned that the values above 1 are just noise (eg. a window with low dS).

One minor point:

Lines 430-433 says "...suggesting production of a defective KKT killer toxin..." This was confusing because there is killing. Why does killing suggest a defective toxin?

**Have all data underlying the figures and results presented in the manuscript been provided?**

Reviewer #1: Yes

PLOS authors have the option to publish the peer review history of their article (what does this mean?). If published, this will include your full peer review and any attached files.

Reviewer #1: No

---

## [Editor Report · Decision Letter 2]

5 Jan 2021

Dear Dr Rowley,

We are pleased to inform you that your manuscript entitled "The Species-specific Acquisition and Diversification of a Family of K1-like Killer Toxins in Budding Yeasts of the Saccharomycotina." has been editorially accepted for publication in PLOS Genetics. Congratulations!

Yours sincerely,

Justin C. Fay

Associate Editor

PLOS Genetics

Hua Tang

Section Editor: Natural Variation

PLOS Genetics

Comments from the reviewers (if applicable):

**Data Deposition**

http://datadryad.org/submit?journalID=pgenetics&manu=PGENETICS-D-20-01491R2

**Press Queries**

---

## [Editor Report · Acceptance letter]

29 Jan 2021

PGENETICS-D-20-01491R2 

The Species-Specific Acquisition and Diversification of a Family of Killer Toxins in Budding Yeasts of the Saccharomycotina. 

Dear Dr Rowley, 

We are pleased to inform you that your manuscript entitled "The Species-Specific Acquisition and Diversification of a Family of Killer Toxins in Budding Yeasts of the Saccharomycotina." has been formally accepted for publication in PLOS Genetics! Your manuscript is now with our production department and you will be notified of the publication date in due course.

With kind regards,

Alice Ellingham

PLOS Genetics

On behalf of:
